# A study of the impact of urban business credit environment on environmental pollution

**Jing Wu**📷*, **Qing Guo**

School of Finance, Harbin University of Commerce, Harbin, China

\* 1813757075@qq.com

**Data Availability Statement:** All relevant data are within the article and its supporting information files. The data in this study relate to 276 cities in China from 2010 to 2021. The data for the

## Abstract

Environmental pollution has become a global concern, so it is critical to find out the elements that influence it. Is the urban business credit environment influencing urban environmental conditions? The study takes the entropy method for calculating the environmental pollution index of 276 prefecture-level cities across China from 2010 to 2021, the CE1 index to describe the urban business credit environment, and a spatial Durbin model to empirically analyze the mechanism and transmission path of the urban business credit environment on environmental pollution. The findings demonstrate that: (1) there is an important positive spatial association between China's urban pollution index and urban business credit index, and that the level of spatial clustering of the two is increasing over time. (2) the urban business credit environment and environmental pollution have an inverted "U" shape relationship and a significant spatial spillover effect, which means that once a certain threshold is reached, improving the urban business credit environment reduces environmental pollution in local and neighboring areas. (3) The influence of the urban business credit environment on environmental deterioration varies by region. The extra heterogeneity test results for the different regions show that the relationship between business credit level and environmental pollution is consistent with the overall situation in the eastern and western regions, the opposite effect was observed in the central region. (4) Changes in the urban business credit environment have an indirect impact on environmental conditions due to the amount of regional financial development and technological innovation potential. To reduce pollution in the region, prefectural-level cities must strengthen the construction of urban business credit systems, strive to create a favorable business credit environment, strengthen the spatial spillover effect, and guide the resource elements of business credit planning to flow into the central and western parts of China reasonably. The study's findings have significant practical implications for modifying the government's development strategy and improving the urban environment.

## 1. Introduction

Strengthening pollution prevention and ecological civilization construction is the primary priority for achieving the optimization of urban economic structure and green sustainable

explanatory variable, the business credit environment, are sourced from the Blue Book of Business Credit Environment Index of Chinese Cities, published by the Business Credit Environment Index Group of Chinese Cities. Other data are derived from publicly accessible resources, including the Statistical Yearbook of China (https://data.cnki.net/yearBook/single?id=N2022110021), the Environmental Statistics Yearbook of China (https://data.cnki.net/yearBook/single?id=N2022030234), the Statistical Yearbook of Science and Technology of China (https://data.cnki.net/yearBook/single?nav=%E6%95%99%E8%82%B2%E7%A7%91%E6%8A%80&id=N2023030111), and the Financial Yearbook of China (https://data.cnki.net/yearBook/single?id=N2022040012), as well as from the EPS data platform and statistical yearbooks of China's prefecture-level cities. Individual city-level data can be accessed from the following respective statistical yearbooks of each city or region: Beijing (http://tjj.beijing.gov.cn/tjsj_31433/), Tianjin (https://stats.tj.gov.cn/tjsj_52032/tjnj/), Hebei (http://www.hetj.gov.cn/hetj/tjsj/jjnj/), Shanxi (http://tjj.shanxi.gov.cn/tjsj/), Inner Mongolia (http://tj.nmg.gov.cn/datashow/pubmgr/publishmanage.htm?m=queryPubData&procode=0003), Liaoning (https://tjj.ln.gov.cn/tjj/tjxx/xxcx/), Jilin (http://tjj.jl.gov.cn/tjsj/tjnj/), Heilongjiang (http://tjj.hlj.gov.cn/tjj/c106782/common_zfxxgk.shtml), Shanghai (https://tjj.sh.gov.cn/tjnj/index.html), Jiangsu (http://tj.jiangsu.gov.cn/col/col87172/index.html), Zhejiang (http://tjj.zj.gov.cn/col/col1525563/index.html), Anhui (http://tjj.ah.gov.cn/ssah/qwfbjd/tjnj/index.html), Fujian (https://tjj.fujian.gov.cn/xxgk/ndsj/), Jiangxi (http://tjj.jiangxi.gov.cn/col/col38595/index.html), Shandong (http://tjj.shandong.gov.cn/col/col6279/index.html), Henan (https://tjj.henan.gov.cn/tjfw/tjcbw/tjnj/), Hubei (http://tjj.hubei.gov.cn/tjsj/sjkscx/tjnj/qstjnj/), Hunan (http://tjj.hunan.gov.cn/hntj/tjsj/tjnj/index.html), Guangdong (http://stats.gd.gov.cn/gdtjnj/), Guangxi (http://tjj.gxzf.gov.cn/tjsj/tjnj/), Hainan (http://stats.hainan.gov.cn/tjj/tjsu/ndsj/), Chongqing (http://tjj.cq.gov.cn/zwgk_233/tjnj/), Sichuan (http://tjj.sc.gov.cn/scstjj/c105855/nj.shtml), Guizhou (http://stjj.guizhou.gov.cn/tjsj_35719/sjcx_35720/gztjnj_40112/tjnj2018/), Yunnan (http://stats.yn.gov.cn/tjsj/tjnj/index.html), Shaanxi (http://tjj.shaanxi.gov.cn/tjsj/ndsj/tjnj/), Gansu (http://tjj.gansu.gov.cn/tjj/c109464/info_disp.shtml), Qinghai (http://tjj.qinghai.gov.cn/tjData/qhtjnj/), and Ningxia (http://nxdata.com.cn/publish.htm?m=getMorePublish&bc=A01&cn=G01). The data used to calculate the distances between prefecture-level cities, based on latitude and longitude coordinates, are from the 1:400,000

development. However, in the context of China's present harsh development model, which prioritizes speed above quality, this has resulted in poor environmental quality, ecological destruction, and a major public health danger situation. At the moment, many scholars are primarily concerned with focusing on government behavior, institutional factors, the legal environment, and other formal institutions that can play a restraining role on urban environmental conditions [1–3], but as China has entered the "credit economy era," the main transaction form of the market has also changed to one dominated by credit transactions. To improve the morality and integrity of the entire society, a social credit system was started to be built in 2003. Scholars have placed a great emphasis on trust as the foundation of the informal system [4–6], Zhang and Chen (2023) found that social trust promotes green innovation and environmental protection by increasing the external environmental pressures faced by firms and enhancing their reputational motivation and environmental awareness [7]. Narain (2022) found that good credit management encourages firms to comply with contracts and commitments, improves the quality of environmental information disclosure to build corporate image and reputation and minimize receivable risks, reduces financial constraints, stimulates innovation, and promotes green innovation to improve the environment [8]. It can be seen that social credit plays an important role in the improvement of the urban environment. Too strong or too weak a degree of credit regulation is detrimental to the quality of green development and must be kept under an appropriate level [9]. As the foundation of the modern credit system and the urban credit system, is also worth paying attention to whether business credit can play an environmental governance effect and promote the sustainable development of China's economy.

At present, research is being conducted to investigate the effects of urban business credit on environmental pollution from both macro and micro dimensions. The macro level is primarily concerned with the environmental influence of the external environment, which includes the institutional, business, and legal environments. For example, Chen et al. (2022) found that implementing the national credit model policies helps to raise the public's understanding of the rules of the law, standardize market order, as well as enhance the effectiveness of the urban green economy [10]. Green credit policy is the best initiative of the government to respond to green development and the construction of a social credit system, Improving green credit levels helps to foster the high-quality growth of the green economy, and raises the standard of high-quality green economy development in one area will raise the quality of high-quality green economy development in adjacent areas [11]. Di et al.(2023) discovered that a credit system based on environmental performance helps to lower emission intensity [12]. A favorable urban business credit climate surely results in an ideal business environment that fosters company innovation, which is more advantageous to the growth of environmentally friendly firms and the productivity of the urban green economy [13]. Yuan et al. (2022) found that The enhancement of the business credit environment fosters the development of the region's financial credit scale, and the resulting financial agglomeration is favorable to the environment up to a certain point [14]. The micro level is primarily developed in terms of credit resource limits and allocation efficiency, as well as firms' green innovation potential. Accessing to and provision of commercial credit not only accelerates the recapitalization of companies, eases financing constraints and reduces company costs, but also brings about advances in corporate technological innovation, improves their production and pollution control technologies, and mitigates the extent of environmental pollution to a certain extent [15–17]. The research on the impact of the business credit environment on environmental pollution is not systematic, the preceding two approaches explored the interaction between the two from the peripheral, to some extent from the macro and micro perspectives, respectively.

topographic database of the National Fundamental Geographic Information System (NFGIS).

**Funding:** The authors declare that no funds, grants, or other support were received during the preparation of this manuscript.

**Competing interests:** The authors have no relevant financial or non-financial interests to disclose.

Although the above literature from the macro and micro perspectives on the relationship between business credit environment and environmental pollution, few studies directly analyze the relationship between urban business credit and environmental pollution, and the first law of geo-economics points out that everything is related to other things, the closer the distance between the two things, the stronger the relevance of this relationship [18]. Although China has a vast territory but uneven distribution of resources, which will inevitably lead to differences in the environmental situation of each region, the spatial spillover effect needs to be considered when exploring the urban business credit environment on environmental pollution. Therefore, this study takes 276 cities in China as an example and deeply analyzes the impact of the urban business credit environment on environmental pollution from both theoretical and empirical perspectives. At the methodological level, spatial econometrics is utilized to explore the spatial spillover effect of the urban business credit environment on environmental pollution. The direct, indirect, and total effects are then further analyzed to provide the necessary theoretical support for the rational formulation and effective implementation of environmental protection policies in China.

This study adds three minor contributions to the literature in this field. (1) By creating a model that places environmental pollution and the urban business credit environment within the same research framework, this paper analyzes the mechanism by which the urban business credit environment influences environmental pollution directly. It also analyzes the direction of the role of the urban business credit environment and the magnitude of its influence on environmental pollution, to uncover the significance of the urban business credit environment. (2) To address the shortcomings, the article builds a spatial measurement model by incorporating the spatial lag of environmental pollution into the explanatory variables of the model. It also examines the effects of the urban business credit environment on adjacent areas from a spatial spillover perspective, expanding our knowledge of spatial correlation and offering resources for further research on inter-regional environmental pollution management. (3) The consequence of the urban business credit environment on local environmental pollution is studied at the national level and in three regions: East, Central, and West. The influence of regional variability in the urban business credit environment on environmental pollution is investigated, and the magnitude of this effect is verified in various regions. (4) Investigate the transmission pathways of urban business credit on environmental pollution through the lens of urban financial development and technology innovation.

The rest of the paper is organized as follows:(1) Section 2 is devoted to the theory and hypotheses. (2) Section 3 details the research design, including model development, description of variables, and data sources and descriptions. (3) Section 4 contains the results of the empirical analyses and a discussion. (4) Section 5 provides conclusions and recommendations.

## 2. Theoretical analysis and research hypothesis

### 2.1 Analysis of the direct impact of urban business credit environment on environmental pollution

Many analysts argue that there is a non-linear relationship between environmental pollution and business credit. Shen, Chen, and Ying (2022) found that rather than being a single promotion relationship, the relationship between the business's credit environment and green total factor production is a reversed "U" shape [19]. The creation of a social credit system depends on the credit system supply process. As a result, the urban business credit environment has the greatest impact on environmental degradation in the two areas listed below. On the one hand, exacerbating the pollution situation, a strengthened urban business credit environment will encourage causing pollution companies to shift their operations abroad out of fear of being placed on the

"blacklist" of serious environmental credit failures and facing cross-sectoral joint disciplinary actions. In the case of an imperfect national credit system and disparities in the strength of environmental credit regulation between regions, firms may choose to relocate to areas with weak green credit regulatory regimes based on the fear of green transformation and being included on the "blacklist" [20], confirming that the choice of relocating production to areas with lagging credit regulation is a major factor in the environmental pollution. Wu et al. (2017) confirm this idea using China's experience, stating that polluting firms tend to relocate from the "east-central-west" due to disparities in environmental credit restrictions [21]. On the other hand, a stronger corporate credit environment may encourage environmentally friendly businesses to progressively adjust to the intensity of environmental restrictions and reduce environmental pollution through scientific and technological advancements. Porter and Linde (1995) note that the establishment of a green credit regulatory regime may force firms to innovate [22]. For starters, under the shared incentive mechanism for creditworthiness, businesses may be required to innovate to be placed on the red list of environmental credits and acquire access to the market. To start, businesses will first follow environmental protection and social responsibility under the joint incentive mechanism for credit compliance to be placed on the environmental red list and receive special treatment concerning marketplace accessibility, accessibility to resources, support for projects, and regulation flexibility. Second, under the "reputation effect," Environmentally friendly enterprises are more likely to be acknowledged throughout society, and this makes it more feasible for them to handle the problem of financing limits through business credit. which in turn leads to an improvement in enterprise innovation ability and production efficiency, which ultimately improves resource utilization rates and reduces waste emissions, thereby reducing environmental pollution. Finally, in the pursuit of sustainable development, enterprises focusing on environmental protection and resource conservation are more likely to be favored by the government, accelerating the digitalization and green transformation of enterprises with the government's blessing, which is also beneficial to environmental protection.

In conclusion, the relationship between urban business credit and environmental pollution is unlikely to be linear. Combining the theory of the inverted "U" shape of the environmental Kuznets curve with the process of China's environmental greening transformation [23], in the early stages of the development of urban business credit, lagging formal finance is still the mainstream, and China's enterprise development is still sloppy. In this regard, reducing environmental pollution is largely dependent on enhancing the business credit environment; during the period of business credit expansion, the "chilling effect" improves the environmental pollution situation. Given the spatial heterogeneity, eastern Chinese cities have a more robust legal system. a more transparent system implementation, and a better business credit environment. While also having an excellent rate of growth in the economy, an abundance of foreign investment, and businesses that are primarily service-oriented, with a smaller impact on environmental pollution [24]. While central and western cities have more outdated systems and significant industries, pollution is more severe. Any type and degree of imbalance will result in a differential influence of the urban business credit environment on environmental degradation. Based on this, Hypothesis 1 is offered.

H1: There is a non-linear link between urban business credit and environmental pollution that rises and then falls, as evidenced by an inverted "U" trend. The urban business credit environment's impact on environmental pollution varies by area.

## 2.2 Analysis of spatial spillover effect of business credit environment on environmental pollution

The term "externality" in microeconomics, which primarily refers to the economic behavior of industrial organizations that will not only have an economic impact on themselves but also

have an impact, either positive or negative, on other industrial organizations, is comparable to the spatial spillover effect. Credit construction has a significant spatial spillover effect because of the sharing and spillover of credit information, the movement of production materials, and the exchange of experiences. This helps to effectively drive and demonstrate credit promotion efforts in underdeveloped regions, encourage full competition in credit construction within each region, form an overall pattern of cooperative incentives for reliability and cooperative penalties for non-compliance at the inter-regional and inter-level levels, and ultimately improve the quality of the region's overall credit environment. Social credit has a significant spatial correlation and spatial spillover effect among provinces. On the one hand, cities with higher levels of urban business credit environment index can both reduce local environmental pollution and, Because of their growth advantages, promote environmental improvement in nearby regions [25]. On the other hand, because the region's urban business credit environment is significantly higher than that of its neighboring regions, there is a large discrepancy in pollutant levels between the two locations, resulting in the "siphon effect". However, there is a substantial difference in environmental pollution degrees between the two regions due to the region's higher urban business credit environment compared to its neighbors. This will result in a "siphon effect" and have an impact on local pollution. Because of this, the following study hypothesis is proposed.

H2: There is a large spatial association between the urban business credit environment and the environmental circumstances of nearby cities. Additionally, there is a significant spatial spillover impact.

## 2.3 Analysis of the mediating effect of business credit environment on environmental pollution

Theoretically, investors' decisions about "whether to invest, where to invest, and how much to invest" are influenced by the legal, institutional, or business credit environment of a region; additionally, investors' actions have a major influence on the technical advancement of the region. According to Han, Pan, and Jin (2023) [13], a favorable business climate fosters an organization's ability for innovation, which is more favorable for the growth of environmental protection businesses and urban green transformation. The development and implementation of financial contracts, however, heavily depend on the protection of the national legal system, as noted by Lopez-de-Silanes et al. (2000) [26]. A robust legal system is necessary to support the country's level of financial development, a region with better creditor protection has larger credit scales and more adequate development of the financial sector, and only when capital conditions are met to meet the demands of the various elements of the enterprise's or region's development process will there be a notable improvement in the environment. will significantly contribute to the enhancement of the environment [14]. It is obvious that the level of regional financial development and innovation in green technology will be influenced by the metropolitan business environment, and that these factors will probably have an impact on regional environmental pollution.

H3: The urban business credit environment affects environmental pollution through both the level of technological innovation and the level of financial development.

## 3. Methodology and description of variables

### 3.1 Methodology

**3.1.1 Benchmark regression model.** The quadratic term of urban business credit environment is added to the following model, which is based on linear benchmark regression, to

confirm the non-linear link between environmental pollution and urban business credit environment:

$$EPI_{it} = \beta_0 + \beta_1 \ln(credit_{it}) + \beta_2 \ln^2(credit_{it}) + \beta_3 X_{it} + \delta_i + \varphi_t + \varepsilon \tag{1}$$

where: In $(credit_{it})$ and $In^2(credit_{it})$ distributions denote the city business credit environment index and the quadratic term of urban business credit environment, $EPI_{it}$ denote the level of environmental pollution, $X_{it}$ denote the relevant control variables, $\delta_i$ and $\varphi_t$ represent city and time fixed effects, respectively, and $\varepsilon_{it}$ are random error terms.

**3.1.2 Spatial Durbin model.** Interregional "siphon effects" and "spillover effects" often lead to the interpenetration and radiation of environmental pollution and economic activities between cities [27]. According to previous studies, environmental pollution has been found to have spatial spillover effects [28]. Ignoring the impact of spatial effects may lead to biased model estimates and a lack of interpretation. This study employs the spatial econometric model for empirical analysis to more objectively measure the spatial spillover features between the environmental pollution index and the urban business credit environment index. The geographic Durbin model provides a more thorough analysis of the relationship between environmental pollution and urban business credit environments because it can quantify variable spillover effects, or direct and indirect effects, both within and between regions.

The first step in establishing a spatial Durbin model is to select a suitable spatial weight matrix, which is generally considered to be of four types: 0–1 spatial weight matrix, geographic distance spatial weight matrix, economic distance spatial weight matrix, and economic-geographic nested spatial weight matrix [29, 30]. The four types of weight matrices are as follows:

$$W_1 = \begin{cases} 1, & i = j \\ 0, & i \neq j \end{cases}$$

If there is a common boundary between city $i$ and city $j$, then note $W_{ij} = 1$, otherwise $W_{ij} = 0$.

$$W_2 = \begin{cases} \dfrac{1}{d_{ij}}, & i \neq j \\ 0, & i = j \end{cases}$$

$d_{ij}$ indicates the geographic distance between city $i$ and city $j$ in terms of the latitude and longitude of the city center.

$$W_3 = \begin{cases} \dfrac{1}{\left| \overline{GDP_i} - \overline{GDP_j} \right|}, & i \neq j \\ 0, & i = j \end{cases}$$

$\overline{GDP_i}$ and $\overline{GDP_j}$ represent the GDP per capita of City $i$ and City $j$, respectively.

$$W_4 = \begin{cases} \varphi^* \left( 1/d_{ij} \right) + (1 - \varphi)^* \left( 1/\left| \overline{pgdp_i} - \overline{pgdp_j} \right| \right) & i \neq j \\ 0 & i = j \end{cases}$$

Among them. $d_{ij}$ indicates the geographic distance between the city $i$ and the city $j$ in terms of the latitude and longitude of the city center. $\overline{pgdp_i}$ denotes the average GDP per capita of the $i$ city during the study period, $\overline{pgdp_j}$ denotes the average GDP per capita of the $j$ city during the

study period, and $\varphi$、 $1-\varphi$ denotes the weights given to geographic and economic distance, $\varphi$ taking the value of 0.5 in this study.

Selecting Moran's I was used to performing the spatial autocorrelation test before estimating the spatial model's parameters. Moran's I have a value between -1 and 1, the stronger the autocorrelation, the closer the values are to -1 and 1. A positive correlation is shown by a Moran's value larger than 0, and a negative correlation is indicated by a value smaller than 0:

$$I = \frac{\sum_{i=1}^{n}\sum_{j=1}^{n}W_{ij}\left(EPI_i - \overline{EPI}\right)\left(EPI_j - \overline{EPI}\right)}{S^2\sum_{i=1}^{n}\sum_{j=1}^{n}W_{ij}} \tag{3}$$

$$I = \frac{\sum_{i=1}^{n}\sum_{j=1}^{n}W_{ij}\left(CREDIT_i - \overline{CREDIT}\right)\left(CREDIT_j - \overline{CREDIT}\right)}{S^2\sum_{i=1}^{n}\sum_{j=1}^{n}W_{ij}} \tag{4}$$

Where, $EPI_i$、 $CREDIT_i$ denotes the specific observed values of the environmental pollution index and urban business credit environment index of a region in the year, respectively, and $n$ is the total number of regional samples.

Consequently, the spatial Durbin model based on model (1) is constructed in this study using the 0–1 spatial weight matrix ($W_1$), and the model's specific form is as follows.:

$$EPI_{it} = \delta\sum_{j=1}^{N}W_{ij}EPI_{it} + \ln\alpha + \beta_1\ln(credit_{it}) + \beta_2\ln^2(credit_{it}) +$$

$$\beta_3 X_{it} + \theta_1\sum_{j=1}^{N}W_{ij}\ln(credit_{it}) + \theta_2\sum_{j=1}^{N}W_{ij}\ln^2(credit_{it}) + \tag{2}$$

$$\theta_3\sum_{j=1}^{N}W_{ij}X_{it} + \mu_i + \lambda_t + \varepsilon_{it}$$

Where $i$ denotes region, $t$ denotes time, $\ln(credit_{it})$ and $\ln^2(credit_{it})$ distributions denote the urban business credit environment index and the quadratic term of city business credit environment and $EPI_{it}$ denote the level of environmental pollution, $X_{it}$ denote the relevant control variables, $W_{ij}$ denotes spatial weights, $\mu_i$ denotes time fixed effects, $\lambda_t$ denotes individual fixed effects, and $\varepsilon_{it}$ is the random error term, Where $W_{ij} EPI_{it}$ denotes the interaction between the dependent variable $EPI_{it}$ and the neighboring unit dependent variable $EPI_{it}$, $\delta$ is the spatial autoregressive coefficient, $\alpha$ is a constant term, and $\beta$ is a fixed unknown parameter.

**3.1.3 Mediated effects model.** In order to verify the two conduction paths of technological innovation level (conduction path I) and financial development level (conduction path II) to explore the mechanism of business credit environment affecting environmental pollution, the following model is established to carry out further analysis:

$$roads_{i,t} = \alpha + \beta\ln(credit_{it}) + \varphi\ln^2(credit_{it}) + \theta X_{i,t} + \varepsilon_{i,t}$$

Where $roads_{i,t}$ characterizes the possible path of business credit environment affecting environmental pollution, $\beta$ characterizes the degree of its influence, $\ln(credit_{it})$ characterizes the

level of business credit environment in the region in the first year, and $X_{i,t}$ represents the control variables.

Combining the above theoretical analysis and modeling process, the research content and methodology are shown in Fig 1.

### 3.2. Description of variables

**3.2.1 Independent variables.** Urban business credit environment (credit): The practice of Chen et al. (2023) to adopt the "China Urban Business Credit Environment Index (CEI)" is cited in this paper [31]. The CEI was created by a collaborative research group comprised of the China Academy of Management Sciences and the China Marketing Association Credit Committee, and it is founded on the concepts of social credit system operation and modern credit management theory. The CEI assesses the regional business credit environment in seven areas: government credit supervision, credit collection system, credit investment, non-compliance and credit violation, enterprise credit management function, enterprise feeling, and honesty education. The CEI is based on public information data from the cities and supports the measurement of each location's business credit environment [32]. In the period when CEI reports were not generated, the indicators of the corporate credit environment were analyzed using the mean value technique with consideration for peer practice.

**3.2.2 Dependent variables.** Environmental Pollution Index (EPI): Borrowing from Fan et al. (2023) [33], this paper selects industrial pollutant emissions of prefecture-level and above cities as an indicator of local environmental pollution control level. The level of environmental pollution (EPI) was calculated using the entropy value approach, which measures the percentage of industrial carbon dioxide, industrial wastewater, and industry dust ultimately released to GDP.

**3.2.3 Control variables.** Other socioeconomic variables can easily influence local environmental pollution; therefore, to reduce the influence of additional factors, the control variables chosen for this work are the degree of economic growth and foreign direct investment., government intervention, environmental regulation, population density, and industrialization level.

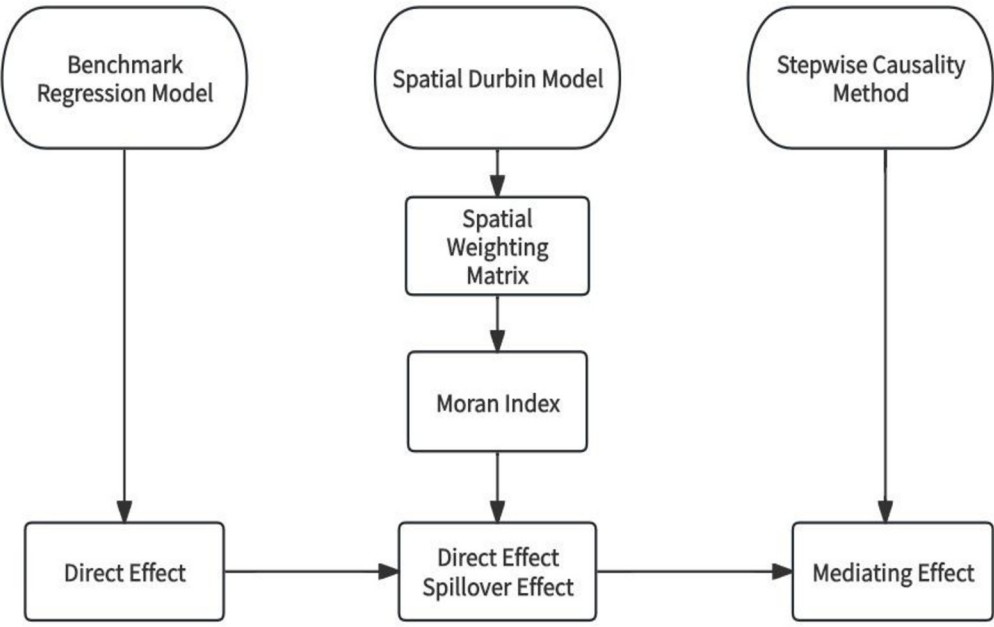

**Fig 1. Methodological framework diagram.**

Economic development level (pgdp): according to the EKC hypothesis, economic development and environmental pollution present a non-linear "inverted U-shaped" relationship [34], measured by the ratio of the GDP of each prefecture-level city to the total population of each prefecture-level city; foreign direct investment (FDI): expressed by the proportion of foreign investment in each prefecture-level city to the GDP of each prefecture-level city [35]; government intervention (gov) is measured by the ratio of the fiscal budget expenditure of each prefecture-level city to the GDP of each prefecture-level city [36]; Environmental regulation (er) is a percentage of the government's investment in environmental pollution control per GDP [37]; The population density (lnPd) is the proportion between the year-end population and the administrative area's land area [38]; industrialization level (ind) is therefore used to express the level of industrialization of a region. The reason was that the characteristics of "high pollution and high emission" in the production process of industry had a certain impact on city air quality [39].

**3.2.4 Intermediary variables.** Financial development level (FDL): expressed as the ratio of each city's GDP at year's end to the total of its loans and loan balances from financial institutions [40].

Technological innovation (GI): A city's level of technology innovation is calculated by measuring the number of green patents and inventions granted each year [41].

## 3.3 Sources of information and descriptive statistics

In this paper, the data of 276 cities in China from 2010 to 2021 are selected, and the data of the explanatory variable business credit environment are from the Blue Book of Business Credit Environment Index of Chinese Cities, which is researched and regularly published by the Business Credit Environment Index Group of Chinese Cities, and the other data are from the Statistical Yearbook of China(https://data.cnki.net/yearBook/single?id=N2022110021), the Environmental Statistics Yearbook of China(https://data.cnki.net/yearBook/single?id=N2022030234.), the Statistical Yearbook of Science and Technology of China(https://data.cnki.net/yearBook/single?nav=%E6%95%99%E8%82%B2%E7%A7%91%E6%8A%80&id=N2023030111),the Financial Yearbook of China(https://data.cnki.net/yearBook/single?id=N2022040012), the EPS data platform, and statistical yearbooks of China's prefecture-level cities. The data that use the latitude and longitude coordinates of each prefecture-level city in China to represent the distance between each prefecture-level city come from the 1:400,000 topographic database of the National Fundamental Geographic Information System (NFGIS). Linear interpolation was used to complete the data for some cities for the years due to insufficient data. For definitions and descriptions of all variables, see specifically Table 1.

As can be observed from the table, there is not much of a difference in the levels of environmental pollution in each location, with the mean value of the explanatory variable EPI being 0.079, the minimum value being 0.001, and the standard deviation being 0.080. With a higher mean and a lower standard deviation, CREDIT is the primary explanatory variable.

## 4. Empirical analysis and discussion

### 4.1 Analysis of spatial autocorrelation of variables

This study uses the global Moran index to perform a global geographic correlation test between the 276 urban environmental pollution index and the urban business credit environment index. Table 2 presents the results of the spatial correlation test. The graphic illustrates that there is a significant regional spillover effect and a strong positive relationship between environmental pollution and urban business credit in China's prefecture-level cities. The Moran index is also significantly bigger than 0. In other words, areas with high levels of pollution in the environment and areas with high levels of environmental pollution gather, as do

**Table 1. Summary statistics.**

| Variables | Obs | Mean | Std.Dev. | Min | Max |
|---|---|---|---|---|---|
| epi | 3312 | 0.079 | 0.08 | 0.001 | 0.815 |
| credit | 3312 | 70.066 | 3.76 | 61.89 | 90.63 |
| GI | 3312 | 518.899 | 1470.154 | 0.001 | 22275 |
| fdl | 3312 | 2.493 | 1.264 | 0.588 | 21.301 |
| pgdp | 3312 | 54064.04 | 37887.58 | 5304 | 932390 |
| fdi | 3312 | 0.017 | 0.018 | 0.014 | 0.21 |
| gov | 3312 | 0.21 | 0.546 | 0.028 | 31.071 |
| er | 3312 | 0.003 | 0.001 | 0.001 | 0.012 |
| ind | 3312 | 46.031 | 11.107 | 0.001 | 89.75 |
| pd | 3312 | 490.669 | 637.521 | 5.43 | 8901.284 |

areas with high levels of credit in the urban sector and areas with high levels of credit in the urban sector, as do areas with low levels of pollution in the environment and areas with low levels of credit in the urban sector. This demonstrates the importance of performing spatial econometric analysis.

In order to further analyze the spatial distribution of the urban business credit environment and environmental pollution in 276 prefecture-level cities, this paper used ArcMap 10.7 to map the spatial distribution of the urban business credit environment and environmental pollution in 2010 and 2021 (shown in Figs 2 and 3).

Fig 2 shows the environmental pollution in 276 cities in 2010 and 2021. This figure indicates the evolution of environmental pollution, and its severity is indicated by the color shades, with darker colors indicating more severe environmental pollution. During the period 2010–2021, the overall pollution situation in the whole of China has decreased. Overall, in terms of spatial distribution, this indicates that pollution is more severe in cities in the central and western parts of the country than in the eastern parts of the country. This may be due to the implementation of the "Western Development" strategy, the Western region vigorously develops the secondary industry, but at the same time, it also has an impact on the environment.

As shown in Fig 3, the urban business credit environment index of 276 prefecture-level cities is divided into three levels. From a spatial point of view, the number of cities with medium and

**Table 2. Global Moran index of environmental pollution and urban business credit environment.**

| Year | Environmental Pollution Index | | Urban Business Credit Environment Index | |
|---|---|---|---|---|
| | Moran's I | P value | Moran's I | P value |
| 2010 | 0.006 | 0.178 | 0.014 | 0.014 |
| 2011 | 0.013 | 0.013 | 0.000 | 0.575 |
| 2012 | 0.012 | 0.006 | 0.005 | 0.236 |
| 2013 | 0.023 | 0.000 | 0.002 | 0.453 |
| 2014 | 0.038 | 0.000 | -0.002 | 0.845 |
| 2015 | 0.028 | 0.000 | -0.006 | 0.711 |
| 2016 | 0.019 | 0.001 | 0.012 | 0.024 |
| 2017 | 0.012 | 0.029 | 0.029 | 0.000 |
| 2018 | -0.003 | 0.903 | 0.059 | 0.000 |
| 2019 | -0.002 | 0.824 | 0.079 | 0.000 |
| 2020 | -0.001 | 0.672 | 0.079 | 0.000 |
| 2021 | 0.002 | 0.435 | 0.079 | 0.000 |

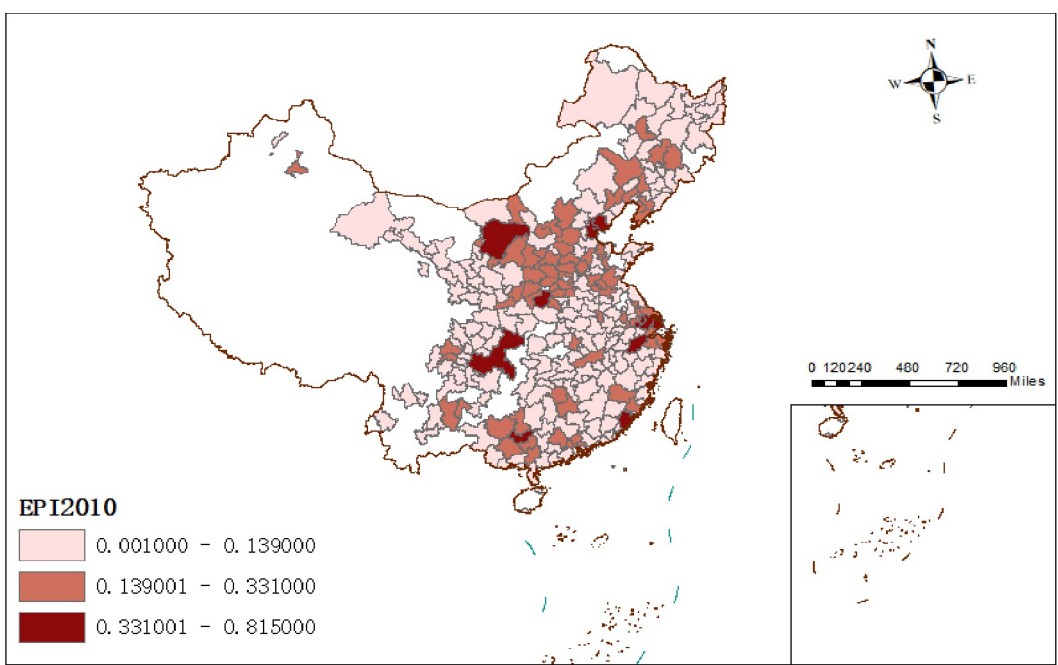

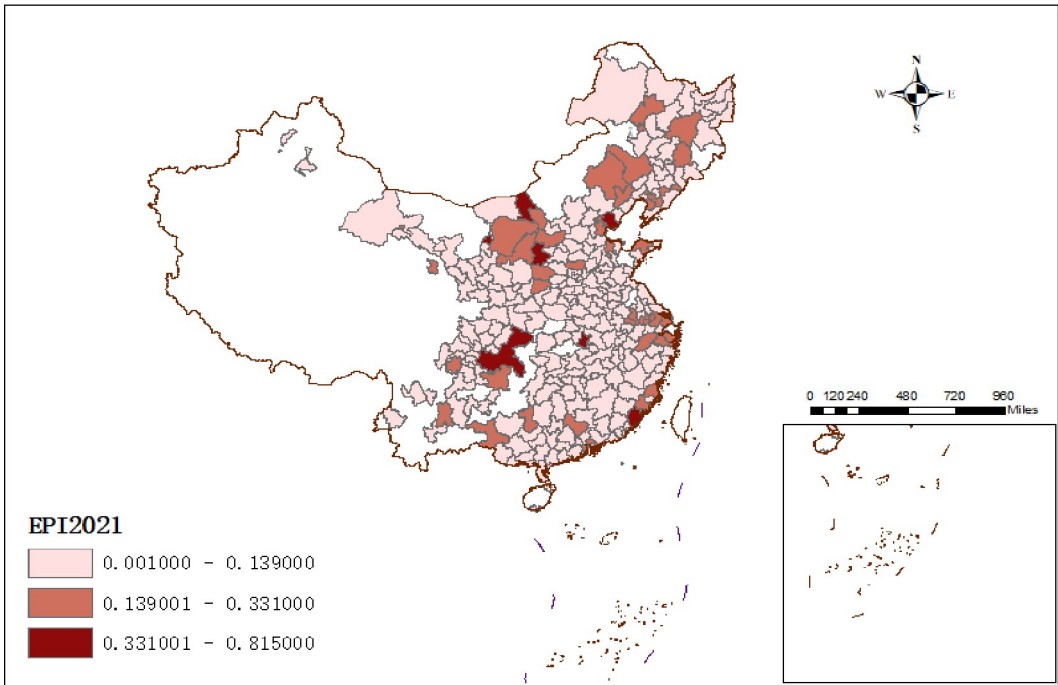

**Fig 2. Spatial characteristics of environmental pollution in 2010 and 2021.** Annotation: Reprinted from [http://bzdt.ch.mnr.gov.cn/] under a CCBY license, with permission from [https://www.tianditu.gov.cn/], original copyright [Map Review Number: GS (2024) 0650].

high levels of urban business credit environment increased significantly from 2010 to 2021, and most of the cities in the Southeast Coastal Region have high levels of business credit, while most of the cities in the Central and Western Regions are at a lower level. This also fully demonstrates that the business credit level of prefecture-level cities in China has regional differences.

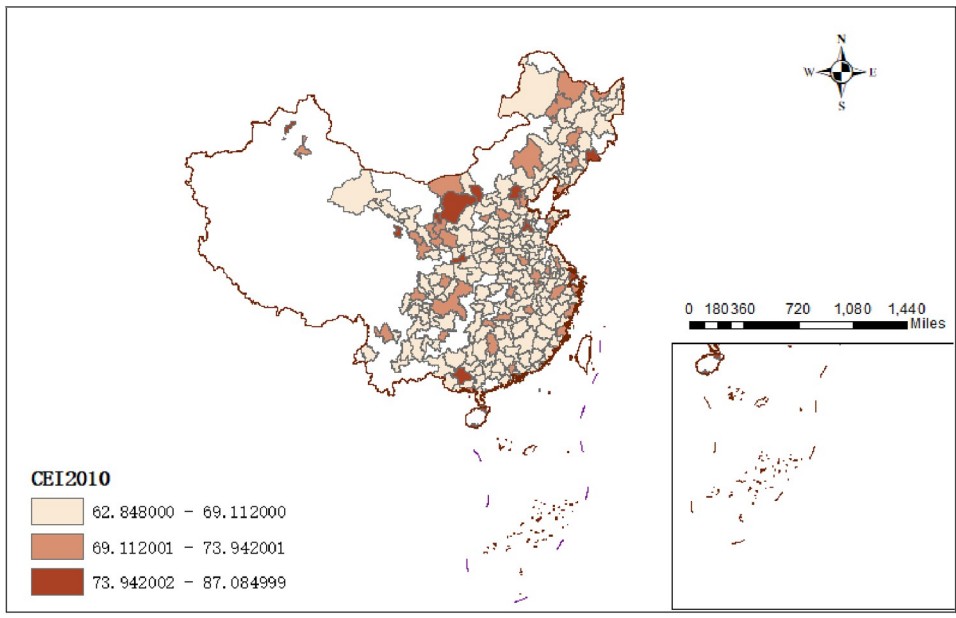

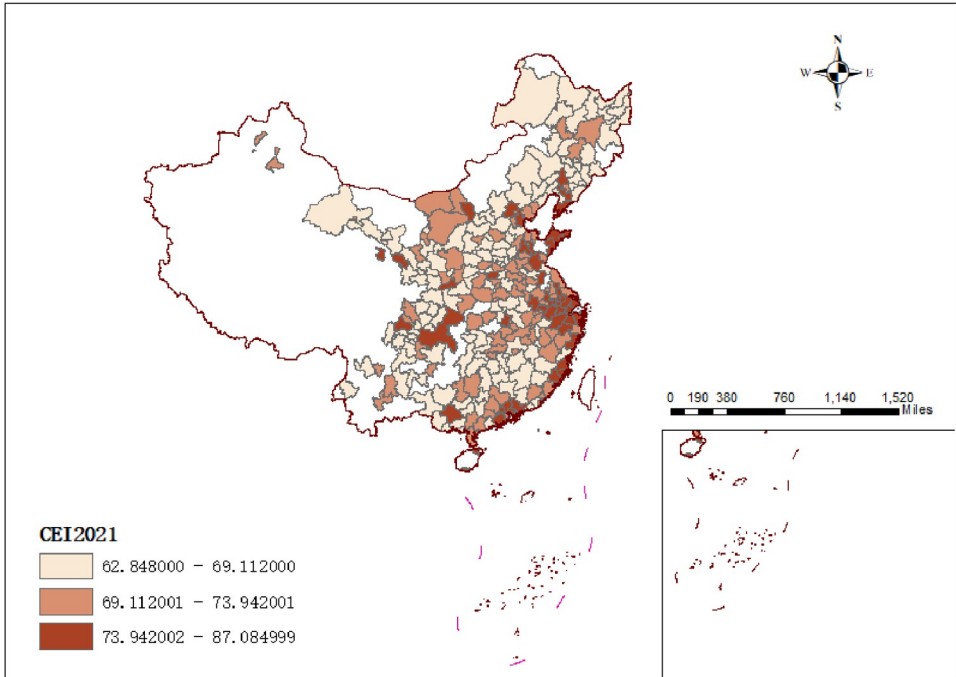

**Fig 3. Spatial characteristics of the urban business credit environment in 2010 and 2021.** Annotation: Reprinted from [http://bzdt.ch.mnr.gov.cn/] under a CCBY license, with permission from [https://www.tianditu.gov.cn/], original copyright [Map Review Number: GS (2024) 0650].

## 4.2 Spatial econometric model selection

Before the analysis, a suitable econometric model is also selected. The spatial weight matrix is first utilized to run a Hausman test, and the findings show that a double fixed effects model should be applied. Next, the choice between the spatial Durbin model (SDM), spatial error model (SEM), and spatial autoregressive model (SAR) is made using the Lagrange multiplier

(LM) and LR tests. The findings in Table 3 demonstrate the importance of all robust LM and LM statistics, suggesting that the spatial Durbin model cannot be refuted and the original hypotheses of "no spatially autocorrelated error term" and "no spatially lagged dependent variable" may be Since the spatial Durbin model cannot be converted into the spatial error model or the spatial autoregressive model, it is used in this paper to investigate the relationship between urban business credit environment and environmental pollution. The spatial Durbin model includes the spatial panel lag and spatial error lag.

## 4.3 Analysis of regression results

In this paper, the ordinary least squares method was first used to regress the urban business credit environment and environmental pollution data of 276 cities in China. Considering the robustness of the results, this paper chooses the SDM model, SAR, and SEM models under the 0–1 spatial weight matrix, and the results are shown in Table 4. where column 1 is the regression result after adding control variables and two-way fixed effects. The 2nd column is the regression result of the SDM model, the 3rd column is the regression result of the SAR model and the 4th column is the regression result of the SEM model.

As can be seen from Table 4, the regression coefficients of the primary term of the urban business credit environment index in the models (1) to (4) have passed at least the positive significance test at the 1% level, and the regression coefficients of the quadratic type have passed at least the negative significance test at the 1% level. As analyzed in the model (1), the increase in the urban business credit environment will exacerbate the environmental pollution before it crosses the inflection point, which is manifested as the increase of the environmental pollution index by 16.0612% for each 1% increase in the urban business credit environment. For every 1% increase in the urban business credit environment, the environmental pollution index will be raised by 16.0612%, but after the urban business credit environment reaches the inflection point value, its rise will reduce the environmental pollution, specifically, for every 1% increase in urban business credit environment, the environmental pollution index will be reduced by 1.9005%. The preliminary results show that there is an inverted "U"-shaped relationship between urban business credit environment and environmental pollution, which first rises and then declines, while the spatial autocorrelation coefficient of the regression results in the model (2)~(4) is significant and negative, which indicates that the environmental pollution has a negative spatial spillover, and this is verified in the model (2)~(5). H1 can be verified.

According to Lesage (2009) [42], the point estimation approach to spatial econometric modeling has drawbacks, and partial differential equations are more effective in explaining the effects of stochastic shocks on variables. Therefore, this paper further decomposes the spatial effect of urban business credit into direct and indirect effects, the direct effect includes the impact of the local urban business credit environment on the level of local environmental

**Table 3. Results of the LM test and LR test.**

| Tests | Categorization | | Statistic | P value |
|---|---|---|---|---|
| LM Tests | Spatial error model(SEM) | LM-Error | 1213.794 | 0.000 |
| | | Robust-LM-Error | 131.899 | 0.000 |
| | Spatial autoregressive model(SAR) | LM-Lag | 1108.816 | 0.000 |
| | | Robust-LM-Lag | 26.920 | 0.000 |
| LR Tests | Spatial autoregressive model(SAR)VS Spatial Durbin model (SDM) | | 19.00 | 0.0082 |
| | Spatial error model(SEM)VS Spatial Durbin model (SDM) | | 18.20 | 0.0111 |

**Table 4. Regression results.**

| | (1) | (2) | (3) | (4) |
|---|---|---|---|---|
| | OLS | SDM | SAR | SEM |
| Main | | | | |
| ln(credit) | 16.0612** | 15.9935*** | 16.0873*** | 16.1734*** |
| | (0.001) | (2.6441) | (2.6444) | (2.6414) |
| ln$^2$ (credit) | -1.9005** | -1.8901*** | -1.9038*** | -1.9143*** |
| | (0.001) | (0.3112) | (0.3113) | (0.3110) |
| lnpgdp | -0.0065 | -0.0098 | -0.0066 | -0.0060 |
| | (0.407) | (0.0058) | (0.0052) | (0.0051) |
| fdi | 0.0098 | -0.2292** | -0.2250** | -0.2225** |
| | (0.517) | (0.0729) | (0.0728) | (0.0727) |
| gov | -0.2266* | -0.0000 | -0.0005 | -0.0005 |
| | (0.015) | (0.0014) | (0.0014) | (0.0014) |
| er | -0.0005 | 2.2498*** | 2.3978*** | 2.4042*** |
| | (0.105) | (0.6195) | (0.6149) | (0.6140) |
| ind | 2.3723** | 0.0005* | 0.0004 | 0.0004 |
| | (0.003) | (0.0002) | (0.0002) | (0.0002) |
| lnpd | 0.0004 | 0.0079 | 0.0099 | 0.0109 |
| | (0.220) | (0.0118) | (0.0110) | (0.0109) |
| Wx | | | | |
| ln(credit) | | 61.4820* | | |
| | | (28.9870) | | |
| ln$^2$ (credit) | | -7.3188* | | |
| | | (3.4161) | | |
| lnpgdp | | 0.0281 | | |
| | | (0.0297) | | |
| fdi | | 0.1379 | | |
| | | (0.7005) | | |
| gov | | 0.0403* | | |
| | | (0.0179) | | |
| er | | 1.2750 | | |
| | | (6.5759) | | |
| ind | | -0.0020 | | |
| | | (0.0014) | | |
| lnpd | | 0.1727 | | |
| | | (0.0902) | | |
| Spatial | | | | |
| rho | | -0.3800** | -0.1814 | -0.2088 |
| | | (0.1459) | (0.1318) | (0.1349) |
| Variance | | | | |
| sigma2_e | | 0.0017*** | 0.0018*** | 0.0018*** |
| | | (0.0000) | (0.0000) | (0.0000) |
| N | 3312 | 3312 | 3312 | 3312 |
| $R^2$ | 0.000 | 0.010 | 0.002 | 0.000 |

pollution and also includes the feedback effect, i.e., after the local urban business credit has an impact on the level of environmental pollution in neighboring areas, the environmental pollution situation in the neighboring areas will in turn have an impact on the development of the

local environmental pollution situation, which is a circular process. The indirect effect indicates the influence of urban business credit in neighboring regions on the development of the environmental pollution control situation. The total effect is the effect of the explanatory variables on all regional explanatory variables, i.e., the urban business credit environment on the environmental pollution situation in the local and all neighboring regions. From the effect decomposition results of the spatial Durbin model, it can be found that the direct, indirect, and total effects of the primary term of the urban business credit environment in Table 5 are all significantly positive, indicating that before crossing the inverted "U" inflection point, the growth of the urban business credit environment will exacerbate the urban environmental pollution, and it has a positive spatial spillover to the neighboring areas, which will increase the pollution of the surrounding environment. Accordingly, H2 is verified.

## 4.4 Robustness check and endogeneity check

**4.4.1 Robustness check.** To ensure the robustness of the above regression results, this paper conducts a robustness test by transforming the spatial weight matrix to further verify the impact of urban business credit level on environmental pollution [43]. Generally speaking, the factors affecting the spatial spillover effect mainly include geographic distance and economic disparity between regions, so this paper adopts the geographic distance weight matrix($W_2$), economic distance weight matrix($W_3$), and economic-geographic nested matrix($W_4$) as the spatial weights as the tool for robustness test. Combined with the goodness-of-fit indicators, Table 6 shows that the coefficient of the quadratic term of the business credit environment is

**Table 5. Direct and indirect effect.**

| VARIABLES | (1) Direct | (2) Indirect | (3) Total |
|---|---|---|---|
| ln(credit) | 15.839*** | 40.129** | 55.968*** |
| | (0.00) | (0.05) | (0.01) |
| $\ln^2$ (credit) | -1.872*** | -4.781** | -6.653*** |
| | (0.00) | (0.05) | (0.01) |
| lnpgdp | -0.010 | 0.024 | 0.015 |
| | (0.13) | (0.35) | (0.55) |
| fdi | -0.225*** | 0.130 | -0.095 |
| | (0.01) | (0.83) | (0.87) |
| gov | -0.000 | 0.029** | 0.028** |
| | (0.81) | (0.02) | (0.03) |
| er | 2.269*** | 1.010 | 3.278 |
| | (0.00) | (0.82) | (0.47) |
| ind | 0.000** | -0.002 | -0.001 |
| | (0.04) | (0.20) | (0.36) |
| lnpd | 0.006 | 0.128** | 0.134** |
| | (0.59) | (0.04) | (0.03) |
| rho | | -0.380*** | |
| | | (0.01) | |
| sigma2_e | | 0.002*** | |
| | | (0.00) | |
| Observations | 3,312 | 3,312 | 3,312 |
| R-squared | 0.010 | 0.010 | 0.010 |
| Number of id | 276 | 276 | 276 |

**Table 6. Robustness check.**

| | (1) Geographic distance matrix | (2) Economic Geography Nested Matrix | (3) economic matrix |
|---|---|---|---|
| **Main** | | | |
| ln(credit) | 12.1400*** | 13.0196*** | 9.9356*** |
| | (2.6878) | (2.6848) | (2.7221) |
| ln$^2$ (credit) | -1.4317*** | -1.5370*** | -1.1730*** |
| | (0.3166) | (0.3162) | (0.3204) |
| **Wx** | | | |
| ln(credit) | 82.2514*** | 40.2680 | 25.7169*** |
| | (22.9946) | (20.9540) | (4.7864) |
| ln$^2$ (credit) | -9.7856*** | -4.8197* | -3.0519*** |
| | (2.7041) | (2.4497) | (0.5633) |
| **Spatial** | | | |
| rho | 0.4817*** | 0.3309** | 0.1841*** |
| | (0.1026) | (0.1032) | (0.0252) |
| **Variance** | | | |
| sigma2_e | 0.0017*** | 0.0017*** | 0.0017*** |
| | (0.0000) | (0.0000) | (0.0000) |
| **Direct** | | | |
| ln(credit) | 12.7623*** | 13.2258*** | 11.0930*** |
| | (2.6800) | (2.6823) | (2.6884) |
| ln$^2$ (credit) | -1.5061*** | -1.5620*** | -1.3107*** |
| | (0.3156) | (0.3158) | (0.3164) |
| **Indirect** | | | |
| ln(credit) | 177.3607** | 67.7996* | 32.5815*** |
| | (57.2753) | (31.8334) | (5.3888) |
| ln$^2$ (credit) | -21.0881** | -8.1045* | -3.8651*** |
| | (6.7526) | (3.7216) | (0.6340) |
| **Total** | | | |
| ln(credit) | 190.1230*** | 81.0254* | 43.6745*** |
| | (57.1530) | (31.6345) | (5.4942) |
| ln$^2$ (credit) | -22.5941*** | -9.6665** | -5.1758*** |
| | (6.7380) | (3.6979) | (0.6465) |
| $N$ | 3312 | 3312 | 3312 |
| $R^2$ | 0.014 | 0.032 | 0.009 |

still significantly negative, and the coefficient of the primary term is significantly positive, which indicates that the urban business credit environment also has an inverted U-shaped relationship with environmental pollution, confirming that the above conclusions are reliable and robust.

**4.4.2 Endogeneity check.** This study addresses the endogeneity problem that may result from two-way causality and omitted variables through two-way fixed-effects modeling of time and individuals. However, it may not be able to fully explain all the relevant variables affecting environmental pollution, such as local policy changes, industrial shifts, or unobserved regional characteristics, and this study can utilize the instrumental variable approach can alleviate the endogeneity problem. For instrumental variable selection, we first refer to Shen, Chen, and Ying (2022) [19], which takes as an instrumental variable whether the 276 cities studied in this

dissertation are the first batch of China's reform and opening-up pioneer cities and special economic zones in the 1980s (IV1), which is taken to be 1 if the city is one of China's reform and opening-up pioneer cities and special economic zones in the 1980s, and is 0 otherwise. The reason for this is that the instrumental variable needs to satisfy correlation and homogeneity. the instrumental variable should be correlated with both the endogenous explanatory variable (urban business credit environment) and uncorrelated with the disturbance term of the explanatory variable (environmental pollution). At the beginning of China's reform and opening up in the 1980s, the arrival of reform and opening up brought about an increase in social mobility in the era when resources were scarce. Against the background of the difficulty of timely follow-up of formal financial support, the scale of business credit expanded dramatically. The special economic zones (SEZs) and the coastal open cities that practiced some of the SEZ policies served as important aspects of China's open-door policy. Therefore, whether the studied city is a special economic zone or one of the first coastal open cities has an important impact on the scale of business credit, which meets the requirement of instrumental variable correlation. Moreover, the reform and opening-up are more than 40 years old, so it does not have a direct impact on the recent environmental pollution in Chinese cities, which meets the requirement of homogeneity. Second, referring to Elbahnasawy (2014) [44], the core explanatory variable lagged one period (L. In(credit)) is chosen as the instrumental variable (IV2). This is because core explanatory variables lagged one period can not only be independent of the random disturbance term in the current period but also reflect the accumulation process of core explanatory variables in the current period [45]. We use the instrumental variables method and two-stage least squares (IV-2SLS) for estimation. The results of IV-2SLS are shown in Table 7. The coefficients of IV1 and IV2 are statistically significant, which indicates that there is a strong correlation between these instrumental and endogenous independent variables. In addition, the F-statistic test value is greater than 10 which indicates the lack of weak instrumental variables among all the selected instrumental variables. Moreover, it has been observed that the p-values of Anderson canon. Corr. LM statistics are all below 0.1, thus negating the initial hypothesis of non-identifiability. Therefore, instrumental variables IV1 and IV2 are valid.

## 4.5 Further analysis

**4.5.1 Analysis of intermediation effects.**   The SDM model was used to validate and analyze the mediating role of financial development and STI from a spatial perspective. The results are shown in Tables 8 and 9. The results show that the influence of urban business credit environment on financial development and science and technology innovation is in the shape of "U". According to this judgment, in the short term, the strengthening of the business credit environment will bring about the reduction of the regional financial development level and scientific and technological innovation ability, which will exacerbate the environmental pollution, and with the further acceleration of the urban business credit environment, the transformation of the financial development consciousness and concepts, as well as the application of advanced technology, will lead to the development of the regional financial industry and the enhancement of the innovation ability will form a constraint on the environmental pollution, and it also confirms that the influence of the business credit environment on the environmental pollution is "inverted U" shape. It also confirms that the inverted U-shaped influence of the business credit environment on environmental pollution is mainly generated through the two conduction paths of technological innovation level and financial development level.

**4.5.2 Analysis of regional heterogeneity.**   Due to the unevenness of regional resource endowment and the different development modes of each city, it may result in the

**Table 7. Endogeneity check.**

| VARIABLES | (1) First Stage In(credit) | (2) First Stage In$^2$ (credit) | (3) Second Stage epi |
|---|---|---|---|
| IV1 | 0.0104*** | 0.0897*** | |
| | (5.08) | (5.14) | |
| IV2 | 0.7584*** | 6.4739*** | |
| | (70.19) | (70.22) | |
| In(credit) | | | 154.6552*** |
| | | | (3,355.13) |
| In$^2$ (credit) | | | -18.1073*** |
| lnpgdp | 0.0011 | 0.0102 | 0.0593*** |
| | (1.10) | (1.21) | (15.87) |
| fdi | 0.1415*** | 1.2113*** | -0.0482 |
| | (4.87) | (4.89) | (-0.43) |
| gov | -0.0011 | -0.0092 | 0.0045 |
| | (-1.32) | (-1.27) | (1.40) |
| er | -0.4680 | -3.7925 | 9.6083*** |
| | (-1.46) | (-1.39) | (7.88) |
| ind | 0.0003*** | 0.0020*** | -0.0025*** |
| | (5.61) | (5.27) | (-14.71) |
| lnpd | 0.0050*** | 0.0426*** | 0.0137*** |
| | (8.68) | (8.74) | (6.19) |
| Anderson canon. corr. LM statistic | | 11.181*** | |
| Cragg-Donald Wald F statistic | | 10.595 | |
| Observations | 3,036 | 3,036 | 3,036 |

heterogeneity of the impact of different cities' business credit levels on environmental pollution. Therefore, after analyzing the role of the urban business credit environment on environmental pollution from the overall level of China, regional heterogeneity is considered, and this paper divides 276 cities into three regions: east, central, and west based on the National Bureau of Statistics' criteria for the division of the three regions. It further discusses the possible differential effects of the urban business credit environment on environmental pollution in different regions of China. The results are shown in columns (1) through (3) of Table 10. In the three parts of the east, central, and west regions, it can be found that the spatial coefficients are first analyzed, and the p-values of the spatial Durbin model in the three regression results are significantly negative, indicating that there is a significant negative spatial effect of the urban business credit environment on environmental pollution. In the east and west, the primary coefficient of the impact of the urban business credit environment on environmental pollution is significantly positive, and the coefficient of the secondary coefficient is significantly negative, indicating that the urban business credit environment and environmental pollution present an inverted "U"-type non-linear relationship, and the indirect effect of the urban business credit environment is significant, indicating that the urban business credit environment and environmental pollution have a spatial spillover effect, but in the east and west, the indirect effect of the urban business credit environment is significant. The indirect effect of the urban business credit environment is significant, indicating that the urban business credit environment and environmental pollution have spatial spillover effects, but the relationship between the two in the central region is "U"-shaped. Hypothesis 1 is verified.

**Table 8. Results of the test of the mediating effect of the level of financial development.**

| VARIABLES | fdl | | EPI | |
|---|---|---|---|---|
| | Main | Wx | Main | Wx |
| In(credit) | -124.555*** | -298.995 | 15.723*** | 57.976** |
| | (0.00) | (0.46) | (0.00) | (0.05) |
| In² (credit) | 14.489*** | 37.712 | -1.858*** | -6.893** |
| | (0.00) | (0.43) | (0.00) | (0.04) |
| fdl | | | -0.003** | -0.011 |
| | | | (0.03) | (0.23) |
| pgdp | -0.000*** | -0.000*** | -0.000 | -0.000 |
| | (0.00) | (0.01) | (0.95) | (0.56) |
| fdi | -3.983*** | -3.495 | -0.253*** | 0.178 |
| | (0.00) | (0.68) | (0.00) | (0.76) |
| gov | -0.022 | 0.085 | 0.000 | 0.040** |
| | (0.29) | (0.74) | (0.92) | (0.03) |
| er | 29.958*** | 120.027 | 2.304*** | 1.076 |
| | (0.00) | (0.20) | (0.00) | (0.87) |
| lnpd | 0.639*** | -3.483** | 0.014 | 0.147 |
| | (0.00) | (0.01) | (0.24) | (0.12) |
| ind | -0.030*** | -0.023 | 0.000 | -0.002 |
| | (0.00) | (0.20) | (0.23) | (0.10) |
| rho | 0.293*** | | -0.409*** | |
| | (0.00) | | (0.01) | |
| sigma2_e | 0.366*** | | 0.002*** | |
| | (0.00) | | (0.00) | |
| Observations | 3,312 | 3,312 | 3,312 | 3,312 |
| R-squared | 0.014 | 0.014 | 0.014 | 0.014 |
| Number of id | 276 | 276 | 276 | 276 |

## 4.6 Discussion

As shown in the regression results in Table 4, there is an inverted U-shaped relationship between urban business credit environment and environmental pollution, that is, before crossing the inflection point, the increase of urban business credit environment will aggravate the environmental pollution, but after reaching the inflection point value, the rise of urban business credit environment will reduce the environmental pollution. It shows that the urban business credit environment has a positive impact on changing environmental conditions. Analyze the reason, when the size of business credit is small, along with its growth China's environmental pollution situation intensifies continuously. At this stage, the business credit environment is a "poison" for the urban environment, because business credit is a kind of public credit with a higher degree of standardization, which is more subject to informal constraints such as reputation and morality, and excessive transaction costs are not conducive to the sustained and healthy operation of enterprises and may reduce the utilization of resources and increase the emission of pollution, which is not conducive to the urban environment. When the business credit environment reaches a critical value, it will play the role of "antidote", which is likely to improve urban pollution. At this time, along with the development of green finance, the formal financial credit support for the green economy continues to increase, but after reaching a certain loan size, under the constraints of legal deposit reserves, deposit absorption, and other factors, the difficulty of formal financial credit increases steeply; as an

**Table 9. Results of the mediation effect test for the level of technological innovation.**

| VARIABLES | ln(GI) | | EPI | |
|---|---|---|---|---|
| | Main | Wx | Main | Wx |
| ln(credit) | -74.732*** | -179.369 | 16.219*** | 61.758** |
| | (0.01) | (0.52) | (0.00) | (0.03) |
| $\ln^2$ (credit) | 8.716*** | 21.263 | -1.916*** | -7.347** |
| | (0.01) | (0.52) | (0.00) | (0.03) |
| ln(GI) | | | 0.003* | 0.005 |
| | | | (0.10) | (0.58) |
| ln(pgdp) | 0.147** | 1.445*** | -0.010* | 0.019 |
| | (0.01) | (0.00) | (0.07) | (0.53) |
| fdi | -0.552 | 7.989 | -0.228*** | 0.033 |
| | (0.45) | (0.26) | (0.00) | (0.96) |
| gov | 0.028* | 0.053 | -0.000 | 0.038** |
| | (0.05) | (0.77) | (0.93) | (0.03) |
| er | 25.311*** | 239.830*** | 2.171*** | 1.027 |
| | (0.00) | (0.00) | (0.00) | (0.88) |
| ind | 0.005** | -0.015 | 0.000** | -0.002 |
| | (0.02) | (0.29) | (0.04) | (0.15) |
| ln(pd) | -0.256** | -0.666 | 0.009 | 0.179** |
| | (0.03) | (0.47) | (0.47) | (0.05) |
| rho | 0.384*** | | -0.382*** | |
| | (0.00) | | (0.01) | |
| sigma2_e | 0.179*** | | 0.002*** | |
| | (0.00) | | (0.00) | |
| Observations | 3,312 | 3,312 | 3,312 | 3,312 |
| R-squared | 0.043 | 0.043 | 0.015 | 0.015 |
| Number of id | 276 | 276 | 276 | 276 |

important backup funding channel outside the formal financial credit, business credit to alleviate the financing constraints highlights the role of business credit to enhance the regional environmental pollution. important role [46].

The effect decomposition results of the spatial Durbin model in Table 5 show that the direct, indirect, and total effects of the primary term of the urban business credit environment are all significantly positive, indicating that the growth of the urban business credit environment will exacerbate the urban environmental pollution before the inverted U-shaped inflection point is crossed, and there is a positive spatial spillover to the neighboring areas, which will exacerbate the surrounding environmental pollution. The positive spatial spillover to the neighboring areas will aggravate the environmental pollution in the surrounding areas. According to the secondary term of urban business credit environment, the direct effect, indirect effect, and total effect are all significant and the coefficient is negative, which indicates that there is an inverted "U" curve relationship between urban business credit environment and environmental pollution, and the effect of urban environmental pollution aggravating environmental pollution is not always increasing, but there is a certain threshold. This result may be because after the urban business credit environment reaches a certain height, the improvement of the urban business credit level is conducive to the local enterprises to alleviate the problem of financing constraints, improve the infrastructure and public services in the urban area, and prompt the enterprises to carry out technological innovation and promote green

development [47, 48]. Under the regional transmission mechanism of the "learning effect", the business credit atmosphere in the region can be radiated to the neighboring regions and improve the environmental conditions of the neighboring cities.

According to the mediation effect regression results in Tables 8 and 9, in the short term, the strengthening of the business credit environment will bring about a reduction in the level of regional financial development and scientific and technological innovation capacity, which will exacerbate the environmental pollution, and with the further acceleration of the business credit environment in the city, the transformation of the financial development awareness and concepts, as well as the application of advanced technology, will lead to the development of the regional financial industry and the enhancement of the capacity of innovation, which will form a constraint on the environmental pollution. One of the reasons may be that of China's lagging financial industry system and system, the rough development in the early stage will bring about environmental degradation, with the formalization of finance, the financial industry development of the green development of the matching degree is gradually enhanced, the business credit environment will reduce environmental pollution. For green innovation, before reaching the critical point, subject to financing constraints, the innovation ability of enterprises is weak, and cannot take into account the protection of the environment, with the construction of the system and the credit joint reward and punishment mechanism is constantly sound, under the role of the reputation effect mechanism, it will accelerate the enterprise business expansion and technological innovation incentives, and the green innovation ability has a positive impact on the reduction of environmental pollution.

Table 10 Regional heterogeneity analysis results can be seen in the East, Central, and West three parts of the region in the urban business credit environment on environmental pollution there is a significant negative spatial effect. And in the east and west of the urban business credit environment on environmental pollution shows an inverted "U" non-linear relationship between the urban business credit environment, the urban business credit environment indirect effect is significant that the urban business credit environment and environmental pollution has a spatial spillover effect, this inverted "U" relationship is not significant in the central cities. This inverted "U"-shaped relationship is not significant in central cities. This may be due to the fact that, compared with the central region, the eastern region has a higher degree of capital abundance for development, and the market operation mechanism is more soundness. Before the business credit environment reaches the critical point, there is a substitution relationship between business credit and lower-cost formal finance, and the financing constraints faced by enterprises are generally weaker. However, after the business credit environment reaches the tipping point, there is a complementary relationship between business credit and formal finance. Enterprises are gradually emphasizing green production, reducing energy consumption, and enhancing environmental protection. In the western region, where the primary industry is the main industry and the level of environmental pollution is low, once business credit is used to obtain funds, it is conducive to prompting enterprises to develop or purchase green technologies to further strengthen the prevention and control of environmental pollution. This provides another way of thinking to better understand the Western development for green transformation. While the central region of the city credit level development is slow, industrial development is still in transition, the degree of marketization is weak, and the central city of industrialization is higher, but brought serious environmental pollution, so the urban business credit environment of urban environmental pollution is not obvious to reduce the effect of enhancement of the role of the lack of. It can be seen that the impact of the urban business credit environment on environmental pollution varies from region to region.

To summarize, the influence of the urban business credit environment on urban environmental pollution has an inverted "U" shape relationship both in the whole and in the region.

**Table 10. Analysis of regional heterogeneity.**

| | (1) | (2) | (3) |
| --- | --- | --- | --- |
| | **East** | **Central** | **West** |
| Main | | | |
| ln(credit) | 19.8868*** | -12.0902* | 18.0445*** |
| | (4.3882) | (6.0479) | (4.5557) |
| $ln^2$ (credit) | -2.3476*** | 1.4287* | -2.1190*** |
| | (0.5137) | (0.7143) | (0.5380) |
| Wx | | | |
| ln(credit) | 136.9219* | -2.7e+02** | -1.5e+02* |
| | (68.3576) | (95.2937) | (56.9006) |
| $ln^2$ (credit) | -16.1348* | 32.3965** | 17.1503* |
| | (7.9900) | (11.2250) | (6.7252) |
| Spatial | | | |
| rho | -0.4165* | -1.1563*** | -0.5404** |
| | (0.2110) | (0.2860) | (0.1888) |
| Variance | | | |
| sigma2_e | 0.0017*** | 0.0018*** | 0.0012*** |
| | (0.0001) | (0.0001) | (0.0001) |
| Direct | | | |
| ln(credit) | 19.3433*** | -9.2515 | 19.9788*** |
| | (4.3583) | (5.9823) | (4.6719) |
| $ln^2$ (credit) | -2.2842*** | 1.0905 | -2.3468*** |
| | (0.5101) | (0.7065) | (0.5515) |
| Indirect | | | |
| ln(credit) | 92.1693* | -1.2e+02*** | -1.1e+02** |
| | (45.7539) | (37.3001) | (38.0085) |
| $ln^2$ (credit) | -10.8644* | 14.8626*** | 12.4860** |
| | (5.3336) | (4.3948) | (4.4894) |
| Total | | | |
| ln(credit) | 111.5126* | -1.3e+02*** | -86.3077* |
| | (46.2054) | (37.6704) | (37.7484) |
| $ln^2$ (credit) | -13.1486* | 15.9531*** | 10.1392* |
| | (5.3862) | (4.4384) | (4.4595) |
| $N$ | 1176 | 1272 | 864 |
| $R^2$ | 0.029 | 0.006 | 0.006 |

# 5. Conclusions and research recommendations

## 5.1 Conclusion

The environmental pollution problem has become the main weakness of the current ecological civilization construction and high-quality economic development, and the informal system of urban business credit may provide an opportunity to solve this problem. Therefore, based on the panel data of 276 prefecture-level cities in China from 2010 to 2021, this paper analyzes the influence of urban business credit environment on environmental pollution and its spatial effect from various angles, such as regional division and spatial effect decomposition, with the help of spatial Durbin model. The main conclusions are as follows: (1) There is a significant positive spatial correlation between China's urban pollution index and urban business credit

index, and the degree of spatial agglomeration of the two has been increasing over time. (2) There is an inverted "U" shape relationship between the urban business credit environment and environmental pollution and a significant spatial spillover effect. (3) The influence of the business credit environment on environmental pollution is regionally heterogeneous. The role of the eastern and western regions is consistent with the overall situation, but in the central region, there is a certain degree of differentiation. (4) The non-linear impact of the business credit environment on environmental pollution is mainly transmitted through two channels: the level of urban financial development and the level of urban innovation.

## 5.2 Policy recommendations

Based on the above conclusions, this paper puts forward the following recommendations:

1. The government should establish a system of environmental credit rules that matches green development. For example, the government should gradually incorporate environmental indicators, such as environmental protection inputs, emission control, and resource use efficiency, into the business credit evaluation system to ensure that the credit evaluation of business entities matches their environmental performance. It can guide enterprises toward a more environmentally friendly and sustainable direction, contributing to the realization of a green economy and sustainable development goals.

2. Implement a dynamic and coordinated linkage development strategy for building a good urban credit system between regions. Strategies such as regional synergy, regional cooperation, and regional integration between regions should further strengthen credit information fusion and sharing, interconnection and mutual checking, organize cross-regional credit construction interaction, promote credit supervision integration, establish and improve cross-regional coordination and co-promotion mechanisms, guide the flow and proliferation of relevant resources to the central and western regions, and achieve the purpose of narrowing the differences in the level of urban business credit between regions, so as to unleash the business credit environment and reduce environmental pollution. dividend of reducing environmental pollution.

3. The government should improve the financial ecological environment and optimize the business credit environment to effectively increase regional financial credit support to promote the improvement of the regional financial development level. Secondly, the local government should use the combination of financial subsidy policy, tax policy, and industrial policy to attract innovative enterprises, further stimulate innovation live, improve the level of urban green technology innovation, and achieve the effect of emission reduction.

It should be noted that this study also has some limitations. First, the limitation of data availability. The official release of the latest data of the City Business Credit Environment Index (CEI Index) is only from 2010 to 2021, so this paper only uses the panel data of 276 cities in China from 2010 to 2021 as the empirical research object. Future research could consider extending the data to prefecture-level cities after 2021, which could help validate the findings and explain any changes in trends over time, draw more specific conclusions, and make more specific policy recommendations. Second, the limitations of the study population. This study focuses only on Chinese cities, which may limit the generalizability of the findings to other contexts. Due to different regulatory, economic, and social conditions, different countries or regions may exhibit different relationships between the business credit environment and environmental pollution. Therefore, future research should expand the study sample to improve the applicability of the findings.

## Supporting information

**S1 Data.**
(XLSX)

## Author Contributions

**Conceptualization:** Jing Wu, Qing Guo.

**Data curation:** Qing Guo.

**Investigation:** Jing Wu.

**Methodology:** Qing Guo.

**Project administration:** Jing Wu.

**Supervision:** Jing Wu.

**Validation:** Jing Wu, Qing Guo.

**Visualization:** Qing Guo.

**Writing – original draft:** Qing Guo.

**Writing – review & editing:** Qing Guo.

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
