## [Decision Letter · Decision Letter 0]

28 Jun 2024

PONE-D-24-15225A study of the impact of urban business credit environment on environmental pollutionPLOS ONE

Dear Dr. Wu,

Thank you for submitting your manuscript to PLOS ONE. After careful consideration, we feel that it has merit but does not fully meet PLOS ONE’s publication criteria as it currently stands. Therefore, we invite you to submit a revised version of the manuscript that addresses the points raised during the review process.

**ACADEMIC EDITOR: **The submission requires further revisions with reference to both the theoretical and quantitative framework.

We look forward to receiving your revised manuscript.

Kind regards,

Stefan Cristian Gherghina, PhD. Habil.

Academic Editor

PLOS ONE

3. We note that your Data Availability Statement is currently as follows: [All relevant data are within the Figure.xlsx]

Reviewers' comments:

Reviewer's Responses to Questions

**Comments to the Author**

1. Is the manuscript technically sound, and do the data support the conclusions?

Reviewer #1: Yes

Reviewer #2: Yes

Reviewer #3: Yes

Reviewer #4: No

2. Has the statistical analysis been performed appropriately and rigorously? 

Reviewer #1: Yes

Reviewer #2: Yes

Reviewer #3: Yes

Reviewer #4: I Don't Know

3. Have the authors made all data underlying the findings in their manuscript fully available?

Reviewer #1: Yes

Reviewer #2: Yes

Reviewer #3: Yes

Reviewer #4: No

4. Is the manuscript presented in an intelligible fashion and written in standard English?

Reviewer #1: No

Reviewer #2: Yes

Reviewer #3: Yes

Reviewer #4: Yes

5. Review Comments to the Author

Reviewer #1: Dear authors

This study empirically analyses the impact of commercial credit on the urban environment by means of a spatial econometric model for China in the period 2010-2021. In doing so, spatial spillover effects and heterogeneity across regions are taken into account. The analysis finds an inverse U-shaped relationship between commercial credit and the urban environment, confirming spatial spillover effects and heterogeneity between regions. Robustness to these results is also tested. With sustainable finance gaining attention as a climate change measure, the results of this study are expected to provide useful policy recommendations. However, the explanations of previous studies, the interpretation of the results and the multiple typos significantly undermine the academic value of this study. Improvements to these are therefore suggested as follows.

The discovery of an inverted U-shaped relationship between business credit and the urban environment is a key result of this study. Nevertheless, in the introduction, the authors refer to previous studies on business credit improving the urban environment, but not to previous studies on its worsening the urban environment. Not only in the Literature Review in Section 2, but also in the Introduction in Section 1, an explanation should be provided regarding business credit worsening the environment and increasing pollution by citing relevant studies.

In Table 6, the indirect effects of In(credit) and In2(credit) are just under three times more effective than their direct effects. In general, the direct effects are considered to have larger values than the indirect effects. A rational explanation should be provided for this result.

The total effect of Model 1 in Table 7 is nearly 190, and the size of the value is outstanding compared to the other results. Why are the value so large?

The spatial weight matrix in 3.2.3 should also be accompanied by a concise explanation, rather than just stating the formula.

In the conclusion, the limitations of this study should be stated.

The paper is very difficult to read due to the typos and formatting flaws as pointed out below. The entire paper should be reviewed to reduce these errors.

In the references, the name of the journal is not mentioned.

In Tables 3 and 4, Chinese words are included and should be corrected into English.

In 3.2.3, the formula contains unnecessary small arrows, which should be deleted.

On page 2, line 27, ‘western regions. however,’ is a period typo. In addition, there is no period typo at the end of this sentence.

On page 8, line 173, ‘Exacerbating’ begins with a capital letter, even though it is in mid-sentence.

On page 26, line 465, the word ‘pollution.’ does not require a period.

On page 32, line 556, ‘Increasing’ starts with a capital letter, despite being in the sentence.

On page 34, line 582, ‘The urban’ is capitalised at the beginning of the sentence, even though it is in the middle of the sentence.

Reviewer #2: Dear author(s),

I had the opportunity to read and revised your article.

I believe your work coherent as the Journal “PLOS ONE”, regarding an interesting topic not fully investigated in the literature until now. However, I have some suggestions and minor concerns that may be improve the quality of your work. You can find additional information in the attached revision file.

With best wishes in the publishing process.

Reviewer #3: Review of manuscript "A study of the impact of urban business credit environment on environmental pollution" (PONE-D-24-15225). The study highlights a positive spatial association between urban pollution and commercial credit indices in China; with an inverted "U" shape relationship indicating that improving the urban business credit environment can reduce pollution once a threshold is reached. Regional variations exist in the impact of business credit on environmental deterioration, emphasizing the need for tailored strategies to strengthen commercial credit systems and promote environmental sustainability in different regions.

• More detailed suggestions and comments are presented as follows:

1. Line 53 – 55: Previous research has demonstrated the significant influence social credit plays in improving the urban environment. Please cite the related research articles for in-depth understanding.

2. Section 1 Introduction: The section is quite lengthy and splitting the section into various sub-sections such as Introduction, Literature Review, Research gap, Scope of the study, and Research questions will be more helpful.

3. In the introduction, you need to connect the current state of research to your paper goals. Please follow the literature review by a clear and concise state of the art analysis. This should clearly show the knowledge gaps identified and link them to your paper goals. Also, please provide the organization of the paper in the last paragraph of the introduction.

4. Section 4 Empirical results and analysis: Would you consider splitting the fourth section, into two parts, Results and Discussion? At present, the length of the results section is too long, which is not conducive to grasping the key points.

5. Section 5 Conclusion and policy implications are too long, and it is recommended that the main conclusions be distilled and streamlined.

6. The sub-section 5.2 Policy Recommendations: It can be included in the separate sub-section into Discussion section.

Reviewer #4: The Manuscript titles A study of the impact of urban business credit environment on environmental pollution(PONE-D-24-15225) tries to investigate the impact based on 276 cities.in China.

following are the comments for the authors.

1. The draft is loosely framed and lacks the discussion section completely

2. The data source and year of data is missing( no citations provided).

3. Spatial mapping of the 276 selected cities can really add value to the paper. The authors should adopt appropriate methods and tools for the same.

4. A Methodology chart is not included in the paper …which is a must.

5. Same theme is repeated several times in the paper and reiterated without proper citations.

6. Only 26 references make the whole research very subjective.

7. This paper needs a complete overhauling.

A major revision of the draft is essential.

6. PLOS authors have the option to publish the peer review history of their article (what does this mean?). If published, this will include your full peer review and any attached files.

Reviewer #1: No

Reviewer #2: No

Reviewer #3: No

Reviewer #4: No

---

## [Author Response · Author response to Decision Letter 0]

30 Jul 2024

Revision Report

Dear Editors and Reviewers:

Thank you for your letter and for the reviewers' comments concerning our manuscript entitled “A study of the impact of urban business credit environment on environmental pollution” (ID: PONE-D-24-15225). Those comments are all valuable and very helpful for revising and improving our paper, as well as the important guiding significance to our research. 

We have studied comments carefully and have made corrections which we hope meet with approval. We tried our best to improve the manuscript and made some changes in the manuscript. These changes will not influence the content and framework of the paper. The summary of corrections and the responses to the reviewer’s comments are as follows. The original comments by reviewers are in blue, and the responses are shown in black. Revised portions are marked in red on the paper.

Your efforts in reviewing the manuscript are highly appreciated.

Once again, we would like to express our sincere gratitude for your thorough review and invaluable feedback. Your detailed comments and suggestions have significantly contributed to enhancing the quality of our manuscript. We greatly appreciate the time and effort you have invested in providing us with constructive guidance. Your insights have been instrumental in refining our work, and we are grateful for your support.

Best Regards.

Yours sincerely,

Jing Wu, Qing Guo

30, July, 2024

Summary of the revision:

Section 1 (Introduction): In the revision of this section, we restructured the lines of the introduction in accordance with the reviewers' comments, dividing the section into subsections such as Introduction, Literature Review, Research Gaps, Scope of the Study, and Organizational Structure. The missing literature was fully added to the original first draft, and the organizational structure of the paper was added to the final section.

Section2 (Theoretical analysis and research hypothesis): In order to avoid the structural confusion caused by the excessive length of Section 4 of the original draft, the modification of this section is mainly to place the theoretical elaboration of the mediated effects model in Section 4 in this section as the third theoretical hypothesis point of this paper.

Section 3 (Methodology and description of variables): There are two main changes in this chapter:(1) A more detailed description of the research methodology of this paper. At the suggestion of reviewer 4, we introduce relevant diagrams and theories to develop a detailed explanation of the spatial Durbin model construction process in the original manuscript. (2) The sources and citations of the data in this paper have been scientifically explained. At the suggestion of two reviewers, we have described the data sources and citations by referring to specific URLs.

Section4 (Empirical analysis and discussion): The main changes in this chapter are the following:(1) In order to see more intuitively the relationship between urban business credit environment and environmental pollution, this section combines the original benchmark regression and spatial Durbin model empirical results together for overall interpretation, which also well avoids the length of the chapter. (2) This section adds an explanation of the spatial and temporal changes in the urban business credit environment and environmental pollution. In order to react to the spatial regional characteristics of the dependent and independent variables. (3) An endogeneity explanation is added to justify this study. (4) In order to avoid the confusing structure of the original manuscript, this revision divides this chapter of the original manuscript into two parts: results and discussion, which makes the line clearer.

Section5 (Conclusions and research recommendations): The main changes in this chapter are the following:

(1) This revision focuses on the streamlining of the main conclusions as well as a more refined approach to initiatives related to policy recommendations. (2) Limitations of the paper have been added.

In addition, the manuscript has been carefully revised considering the language and grammar problems, including the sentences that are convoluted and hard to follow.

Respond to the editor's comments

1. Comment:

Dear Dr. Wu,

Thank you for submitting your manuscript to PLOS ONE. After careful consideration, we feel that it has merit but does not fully meet PLOS ONE’s publication criteria as it currently stands. Therefore, we invite you to submit a revised version of the manuscript that addresses the points raised during the review process. 

1. Reply: Thank you for your thorough review and valuable suggestions for my manuscript. Your meticulous examination and professional insights have greatly contributed to the improvement and enhancement of this paper in multiple aspects. Your recommendations have not only helped us delve deeper into the theoretical and methodological aspects but also made the structure and presentation of the article more rigorous and clearer.

I deeply appreciate the time and effort you have dedicated to this review, and I am pleased to have benefited from your feedback. Your suggestions have been crucial in elevating the quality and academic contribution of the paper. We hope that these improvements will enable the manuscript to better meet the journal's standards and provide valuable insights for research in the related field.

Once again, thank you for your careful review and invaluable feedback. I hope this manuscript will receive your approval and be accepted for publication.

2. Comment: The submission requires further revisions with reference to both the theoretical and quantitative framework.

2. Reply: Thank you for reviewing our paper and providing valuable suggestions. I have substantially revised the original manuscript in terms of theory as well as quantity, in accordance with the comments of the reviewers.

3. Comment: Please ensure that your manuscript meets PLOS ONE's style requirements, including those for file naming.

3. Reply: Thank you for providing the style template, we have strictly followed the template provisions of the original manuscript to modify the hope that it can meet your requirements. 

4. Comment: Please provide a complete Data Availability Statement in the submission form, ensuring you include all necessary access information or a reason for why you are unable to make your data freely accessible. If your research concerns only data provided within your submission, please write "All data are in the manuscript and/or supporting information files" as your Data Availability Statement.

4. Reply: Thank you for reviewing our paper and providing valuable suggestions. We highly appreciate your feedback and have made the necessary revisions accordingly. After your suggestion, we have made a data availability statement at the time of submitting the data, which includes the raw data, the source of the data and the citation. Thus, the scientific validity of this study is explained. 

Respond to the Reviewer#1's comments

1. Comment: This study empirically analyses the impact of commercial credit on the urban environment by means of a spatial econometric model for China in the period 2010-2021. In doing so, spatial spillover effects and heterogeneity across regions are taken into account. The analysis finds an inverse U-shaped relationship between commercial credit and the urban environment, confirming spatial spillover effects and heterogeneity between regions. Robustness to these results is also tested. With sustainable finance gaining attention as a climate change measure, the results of this study are expected to provide useful policy recommendations. However, the explanations of previous studies, the interpretation of the results, and the multiple typos significantly undermine the academic value of this study. Improvements to these are therefore suggested as follows.

1. Reply: Thank you for your thorough review and valuable suggestions for my manuscript. Your meticulous examination and professional insights have greatly contributed to the improvement and enhancement of this paper in multiple aspects. Your recommendations have not only helped us delve deeper into the theoretical and methodological aspects but also made the structure and presentation of the article more rigorous and clearer.

I deeply appreciate the time and effort you have dedicated to this review, and I am pleased to have benefited from your feedback. Your suggestions have been crucial in elevating the quality and academic contribution of the paper. We hope that these improvements will enable the manuscript to better meet the journal's standards and provide valuable insights for research in the related field.

Once again, thank you for your careful review and invaluable feedback. I hope this manuscript will receive your approval and be accepted for publication.

2. Comment: The discovery of an inverted U-shaped relationship between business credit and the urban environment is a key result of this study. Nevertheless, in the introduction, the authors refer to previous studies on business credit improving the urban environment, but not to previous studies on its worsening the urban environment. Not only in the Literature Review in Section 2 but also in the Introduction in Section 1, an explanation should be provided regarding business credit worsening the environment and increasing pollution by citing relevant studies.

2. Reply: Thank you for reviewing our paper and providing valuable suggestions. We highly appreciate your feedback and have made the necessary revisions accordingly. At your suggestion, we have added the literature research on credit to the deterioration of the environment, but in view of the main purpose of the introduction is to draw out the relationship between the urban commercial credit environment and environmental pollution, for the urban commercial credit environment for the positive and negative relationship between the environmental pollution of the relevant literature research and theoretical elaboration is concentrated in the second section of this paper, the first section of the introduction mainly plays a role in drawing out the theme. 

We have specifically added the following content:

Zhang and Chen found that social trust promotes green innovation and environmental protection by increasing the external environmental pressures faced by firms and enhancing their reputational motivation and environmental awareness. Narain found that good credit management encourages firms to comply with contracts and commitments, improves the quality of environmental information disclosure to build corporate image and reputation and minimize receivable risks, reduces financial constraints, stimulates innovation, and promotes green innovation to improve the environment .it can be seen that social credit plays an important role in the improvement of the urban environment. However, too strong or too weak a degree of credit regulation is detrimental to the quality of green development and must be kept under an appropriate level. (On page 3, lines 50-61).

3. Comment: In Table 6, the indirect effects of In(credit) and In2(credit) are just under three times more effective than their direct effects. In general, the direct effects are considered to have larger values than the indirect effects. A rational explanation should be provided for this result.

3. Reply: For the spatial Durbin model in the indirect effect is more than three times the direct effect of the problem, I think it is a normal empirical phenomenon, the reasons are as follows: (1) the direct effect reflects the degree of influence of the explanatory variables on the local, the indirect effect reflects the influence of the explanatory variables on the surrounding areas, is a kind of spatial spillover effect of the realization, the coefficient of the indirect effect is much larger than the coefficient of the direct effect of the problem, which means the coefficient is larger than the coefficient of the direct effect of the problem. The influence of a region's urban business credit environment on environmental pollution in its neighboring regions is much larger than its influence on environmental pollution in the region, which means that the spatial spillover effect is more obvious. (2) It is normal that the coefficient of indirect effect is larger, generally it may be manifested because of the problem of magnitude, because the variables are weighted by spatial weights, often the magnitude will be smaller, and when the explanatory variables themselves are smaller, they tend to present larger coefficient results. This can also lead to much larger coefficients for indirect effects than for direct effects. (On page 26, line 495 table 5).

4. Comment: The total effect of Model 1 in Table 7 is nearly 190, and the size of the value is outstanding compared to the other results. Why are the value so large?

4. Reply: Thank you for reviewing our paper and providing valuable suggestions. We highly appreciate your feedback and have made the necessary revisions accordingly. The results of spatial Durbin modeling using geographic distance weight matrix, economic-geographic nested matrix, and economic distance weight matrix as spatial weights, respectively, are presented in Table 6. Among them, model (1) is the result under the use of a geographic distance matrix, and its total effect appears larger compared to the other two models, which I think is normal and fully consistent with the empirical requirements. Selecting the spatial weight matrix is the first step of spatial measurement, and different choices of spatial weight matrix will bring different weighting effects on the explanatory variables, which will lead to different total effects, and the reason for the large effect of this model (1) may be that the weight matrix constructed by utilizing the factor of the distance between the cities has a greater influence on the model, but it still has not changed the characteristics of the relationship between the independent variables and the dependent variable. So it is said that the difference in the choice of weight matrix will bring different effects. However, it also shows that even if the spatial weight matrix changes, the effect of the model remains unchanged, and the use of the model to explore the relationship between the two remains robust. (On page 28, line 510 table 6). Once again, thank you for your valuable feedback. We look forward to your further comments.

5. Comment: The spatial weight matrix in 3.2.3 should also be accompanied by a concise explanation, rather than just stating the formula. In 3.2.3, the formula contains unnecessary small arrows, which should be deleted.

5. Reply: We have briefly explained the spatial weight matrix of 3.2.3 in the revised version. We have specifically added the following content:

If there is a common boundary between city and city , then note =1, otherwise =0. (On page 13, lines 272-274).

6. Comment: In the conclusion, the limitations of this study should be stated.

6. Reply: Thank you for reviewing our paper and providing valuable suggestions. We highly appreciate your feedback and have made the necessary revisions accordingly. After combing through the article, we add two limitations of this study after the policy recommendations in Section 5. Once again, thank you for your valuable feedback. We look forward to your further comments.

We have specifically added the following content:

It should be noted that this study also has some limitations. First, the limitation of data availability. The official release of the latest data of the City Business Credit Environment Index (CEI Index) is only from 2010 to 2021, so this paper only uses the panel data of 276 cities in China from 2010 to 2021 as the empirical research object. Future research could consider extending the data to prefecture-level cities after 2021, which could help validate the findings and explain any changes in trends over time, draw more specific conclusions, and make more specific policy recommendations. Second, the limitations of the study population. This study focuses only on Chinese citie

---

## [Decision Letter · Decision Letter 1]

4 Sep 2024

A study of the impact of urban business credit environment on environmental pollution

PONE-D-24-15225R1

Dear Dr. Wu,

We’re pleased to inform you that your manuscript has been judged scientifically suitable for publication and will be formally accepted for publication once it meets all outstanding technical requirements.

Kind regards,

Stefan Cristian Gherghina, PhD. Habil.

Academic Editor

PLOS ONE

Additional Editor Comments (optional):

Reviewers' comments:

Reviewer's Responses to Questions

**Comments to the Author**

1. If the authors have adequately addressed your comments raised in a previous round of review and you feel that this manuscript is now acceptable for publication, you may indicate that here to bypass the “Comments to the Author” section, enter your conflict of interest statement in the “Confidential to Editor” section, and submit your "Accept" recommendation.

Reviewer #1: All comments have been addressed

Reviewer #3: All comments have been addressed

2. Is the manuscript technically sound, and do the data support the conclusions?

Reviewer #1: Yes

Reviewer #3: Yes

3. Has the statistical analysis been performed appropriately and rigorously? 

Reviewer #1: Yes

Reviewer #3: Yes

4. Have the authors made all data underlying the findings in their manuscript fully available?

Reviewer #1: Yes

Reviewer #3: (No Response)

5. Is the manuscript presented in an intelligible fashion and written in standard English?

Reviewer #1: Yes

Reviewer #3: Yes

6. Review Comments to the Author

Reviewer #1: Dear author.

This paper appears to have been appropriately revised in line with the reviewers' comments.

I would therefore like to leave further decisions to the Editor-in-Chief.

Sincerely.

Reviewer #3: I am satisfied with the revisions made by the authors and I am happy to recommend this manuscript for publication in PLOS ONE.

7. PLOS authors have the option to publish the peer review history of their article (what does this mean?). If published, this will include your full peer review and any attached files.

Reviewer #1: No

Reviewer #3: No

---

## [Editor Report · Acceptance letter]

11 Sep 2024

PONE-D-24-15225R1 

PLOS ONE

Dear Dr. Wu, 

I'm pleased to inform you that your manuscript has been deemed suitable for publication in PLOS ONE. Congratulations! Your manuscript is now being handed over to our production team.

Kind regards, 

on behalf of

Dr. Stefan Cristian Gherghina 

Academic Editor

PLOS ONE